# ECP-IEM: Enhancing seasonal crop productivity with deep integrated models

**Ghulam Mustafa**[1]***, Muhammad Ali Moazzam**[1]**, Asif Nawaz**[1]**, Tariq Ali**[1]**, Deema Mohammed Alsekait**[2]**, Ahmed Saleh Alattas**[3]**, Diaa Salama AbdElminaam**[4,5,6]

**1** University Institute of Information Technology, PMAS–Arid Agriculture University, Rawalpindi, Pakistan, **2** Department of Information Technology, College of Computer and Information Sciences, Princess Nourah Bint Abdulrahman University, Riyadh, Saudi Arabia, **3** Information Science Department, King Abdulaziz University, Jeddah, Saudi Arabia, **4** Information Systems Department, Faculty of Computers and Artificial Intelligence, Benha University, Benha, Egypt, **5** MEU Research Unit, Middle East University, Amman, Jordan, **6** Jadara Research Center, Jadara University, Irbid, Jordan

* gmustafa@uaar.edu.pk

**Data Availability Statement:** The relevant data supporting this paper can be found at https://www.kaggle.com/datasets/patelris/crop-yield-prediction-dataset and https://widgets.figshare.com/articles/27612663/embed?show_title=1.

## Abstract

Accurate crop yield forecasting is vital for ensuring food security and making informed decisions. With the increasing population and global warming, addressing food security has become a priority, so accurate yield forecasting is very important. Artificial Intelligence (AI) has increased the yield accuracy significantly. The existing Machine Learning (ML) methods are using statistical measures as regression, correlation and chi square test for predicting crop yield, all such model's leads to low accuracy when the number of factors (variables) such as the weather and soil conditions, the wind, fertilizer quantity, and the seed quality and climate are increased. The proposed methodology consists of different stages, like Data Collection, Preprocessing, Feature Extraction with Support Vector Machine (SVM), correlation with Normalized Google Distance (NGD), feature ranking with rising star. This study combines Bidirectional Gated Recurrent Unit (Bi-GRU) and Time Series CNN to predict crop yield and then recommendation for further improvement. The proposed model showed very good results in all datasets and showed significant improvement compared to baseline models. The ECP-IEM achieved an accuracy 96.34%, precision 94.56% and recall 95.23% on different datasets. Moreover, the proposed model was also evaluated based on MAE, MSE, and RMSE, which produced values of 0.191, 0.0674, and 0.238, respectively. This will help in improving production of crops by giving an early look about the yield of crops which will than help the farmer in improving the crops yield.

## 1. Introduction

Seasonal crop yield prediction is a crucial aspect of agricultural planning and management [1]. It involves estimating the amount of crop that will be produced in a specific season, which is essential for ensuring food security, optimizing resource allocation, and stabilizing market prices [2]. With accurate predictions, farmers can make informed decisions. For example, knowing the expected yield allows farmers to plan their planting schedules and select

**Funding:** Princess Nourah bint Abdulrahman University PNURSP2024R435 Deema Mohammed Alsekait.

**Competing interests:** The authors have declared that no competing interests exist.

appropriate crop varieties, ensuring they maximize productivity and avoid overproduction or shortages [3]. Additionally, accurate yield forecasts help farmers allocate resources more efficiently, such as water and fertilizers, ensuring they are used where they will have the most significant impact [4]. This optimization reduces waste and costs, enhancing overall farm profitability and sustainability.

Over the past few decades, various initiatives have been launched to combat global hunger and sustain the growing population. Despite a substantial rise in agricultural productivity over the last half-century, nearly 800 million individuals still suffer from inadequate food access and hunger [5]. Consequently, the United Nations' 2030 Agenda for Sustainable Development has prioritized the fight against hunger and the enhancement of food security as its foremost goal [6]. For all those engaged in the production and trading stage of agriculture, predicting the crop's potential yield is an important milestone. In the context of global food crises, the importance of accurate crop yield prediction cannot be overstated. With the world's population continually growing, the demand for food is increasing, putting pressure on agricultural systems to produce more with fewer resources. Natural disasters, climate change, and geopolitical conflicts further exacerbate food insecurity, making it imperative to have robust systems in place for predicting crop yields. Accurate yield predictions enable governments and organizations to plan for potential food shortages, distribute resources effectively, and implement timely interventions to prevent crises [7].

Crop output forecasting may prove to be a helpful instrument for helping prepare and carry out tasks with greater knowledge. This makes forecasting crop output a challenging task that needs to be resolved. Crop yield levels are influenced by a number of variables, including temperature and soil conditions, fertilizer use, and seed variety also establishing the physical characteristics of the organism [8]. It has been possible to estimate agricultural yields using a variety of crop simulation and yield estimation models. Using the factors mentioned above, AI can be used to estimate crop yields more accurately. A branch of AI known as ML has recently been used extensively for crop output prediction due to its capacity to find non-linear rules and patterns in huge datasets that come from various sources [9].

Traditional methods as depicted in Fig 1, of crop yield improvement have relied heavily on statistical models and expert knowledge. These methods typically involve the use of historical yield data, weather conditions, soil characteristics, and other agronomic factors to develop regression models [10]. While these models can provide useful insights, they often fall short in capturing the complex and dynamic nature of agricultural systems. Factors such as changing weather patterns, pest infestations, and technological advancements in farming practices are challenging to incorporate accurately into traditional models [11]. The limitations of traditional crop yield prediction methods are significant. Firstly, these methods often lack the ability to generalize across different regions and crop types, leading to varying degrees of accuracy. Secondly, traditional models are usually static, meaning they cannot adapt to real-time changes in environmental conditions [12]. This rigidity makes them less effective in dealing with the uncertainties and variability inherent in agriculture. Additionally, the reliance on historical data may not always reflect current and future trends, especially in the face of rapid climate change [13].

The rise of machine learning (ML) has revolutionized the approach to predicting crop yields. ML techniques are capable of analyzing extensive datasets, uncovering trends, and generating predictions with greater precision than conventional approaches [14]. Approaches such as support vector machines (SVM), random forests, and gradient boosting have been utilized in forecasting crop yields, employing varied datasets including meteorological data, soil characteristics, and satellite imagery. These methods can learn from past data and adjust to new information, making them both adaptable and reliable [15]. Moreover, deep learning, a

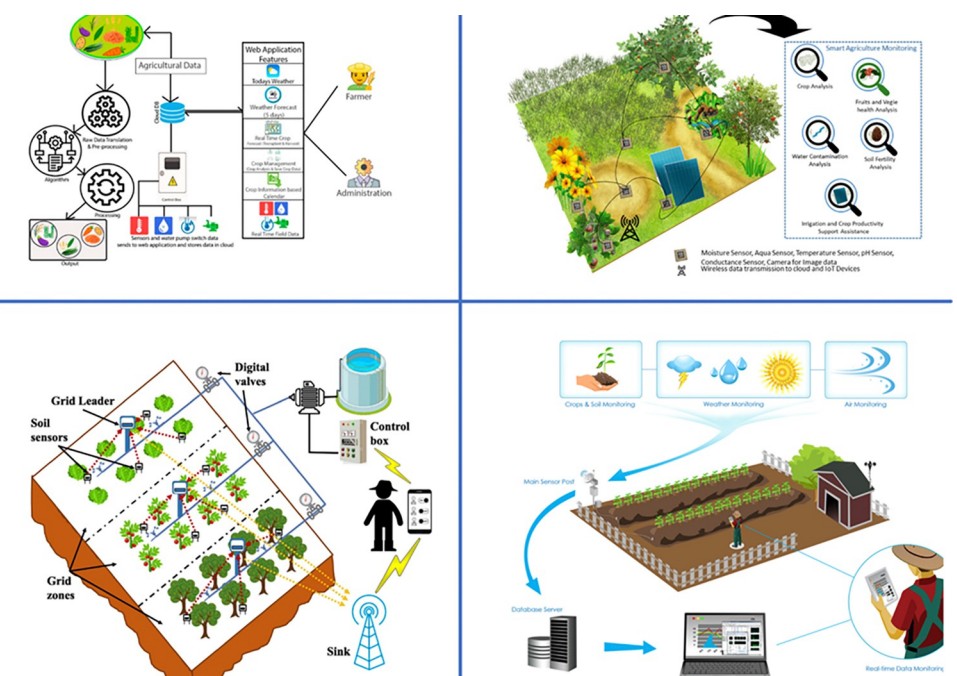

**Fig 1. The work flow of traditional crop yield improving methods.**

branch of machine learning, has significantly improved the performance of crop yield prediction models. Deep learning methods, including convolutional neural networks (CNNs) and recurrent neural networks (RNNs), can manage complex, high-dimensional data and discern intricate correlations between variables. For example, CNNs are particularly adept at processing spatial information from satellite images, while RNNs excel at modeling temporal sequences such as weather trends. The capability of deep learning models to develop hierarchical data representations renders them potent tools for precise yield forecasting [16].

Although machine learning and deep learning techniques offer numerous benefits, they also come with certain limitations. A significant challenge is the necessity for extensive, high-quality datasets to train these models effectively. In many areas, such data can be limited or unreliable. Furthermore, these models often demand substantial computational power, requiring considerable resources for both training and deployment. Another issue is the potential for overfitting, where a model may excel with training data but perform poorly on new, unseen data. Additionally, the opaque nature of many deep learning models can complicate the interpretation of results and understanding of the underlying decision-making process. To address the limitations of existing methods, this research proposes an deep integrated model for improving seasonal crop yield prediction. This model integrates multiple advanced techniques to enhance accuracy and robustness. The key steps of our approach include data collection, preprocessing, SVM-based feature extraction, NGD-based feature correlation, rising star-based feature ranking and an deep integrated model composed of Bi-GRU and time series CNN.

## 1.1 Research contribution

The key contribution of the proposed work is summarized as follows:

- Provides precise guidance on planting, irrigation, and fertilization schedules to boost crop health and yield, while promoting efficient resource use to minimize waste and

environmental impact. Additionally, it supports farmers' economic stability by informing market strategies and investment decisions with accurate yield predictions.

- Introducing advanced feature extraction, correlation, and ranking techniques with a deep integrated model composed of Bi-GRU and time series CNN, resulting in highly accurate and robust crop yield predictions that enhance decision-making and resource optimization for farmers.

- The proposed model was evaluated against baseline approaches, and comparative analysis indicates that it achieved superior results.

The rest of the paper as organized as follows: Section 2 discusses the current literature related to the crop yield improvement techniques, section 3 provides the core methodology of the proposed work, section 4 gives detail about experimental evaluations and results. Section 5 detailed the conclusion and future work direction.

## 2. Literature review

Numerous researches have been found in the field of agriculture and yield improvement such as Paudel et al. [17] created a ML framework that combined crop modeling and ML to predict large-scale crop yield, focusing on accuracy, flexibility, and reusability while avoiding data leakage. The system utilized data from MCYFS database and adapted well to different crops, predicting yield for five crops in three countries and comparing favorably to a basic method in early season predictions, though some crops had higher errors. Future improvements could involve newer data sources, refined features, and diverse ML algorithms. Qiao et al. [18] introduced KSTAGE, which uses past experience for spatial-temporal interactions, employing a 3D CNN and KTMA for attention weights. A Spatial Attention Graph Network merges spatial features, enhancing county-level yield prediction in China and the CONUS. The approach highlights the value of self-attention and graph representation for intricate correlations, with future plans to integrate external factors like weather and soil conditions for better accuracy.

Liu et al. [19] explored wheat yield prediction in Indo-Gangetic Plains via satellite data and various methods, favoring ML and DL over linear regression, with SVR outperforming LSTM. High-resolution data excelled, especially during 2010 extreme weather, while traditional indices matched in 2018. Enhanced satellite data could enhance predictions and aid forecasting in data-scarce regions. Meroni et al. [20] explored the use of small data and ML algorithms for monthly yield prediction during the growing season. They developed an automated ML process that effectively selected features and models. In their case study on Algeria, accurate national yield forecasts were achieved for barley, soft wheat, and durum wheat. ML models consistently outperformed simple benchmarks, even during poor yield years, providing valuable early warning capabilities. While variations in accuracy between models were observed, suitable model calibration and selection ensured benchmark models outperformed a significant portion of tested ML models. It was noted that extensive calibration is necessary to achieve superior performance over simple benchmarks when working with limited data.

The work of Abbaszadeh et al. [21] introduced a framework using Bayesian Model Averaging (BMA) and Copula functions, integrating outputs from multiple deep neural networks like 3DCNN and ConvLSTM for probabilistic soybean crop yield estimation in U.S. counties. The approach outperforms individual networks when considering model uncertainties. The framework's adaptability extends to other model outputs and crops, with future plans to apply it to maize and corn yield predictions across additional U.S. states. Zhu et al. [22] also introduced a new DL adaptive crop model (DACM) for precise large-area yield estimation, emphasizing adaptive learning of spatial crop growth patterns. DACM's stability analysis and attention

values demonstrated its ability to learn and adapt to spatial crop development, providing a valuable approach for accurate and interpretable large-scale yield prediction.

Oikonomidis et al. [23] developed DL models for soybean crop yield prediction, with the hybrid CNN-DNN model achieving the best performance (RMSE 0.266, R2 0.87). XGBoost showed the second-best performance and faster runtime. The authors propose investigating hybrids of XGBoost and DL methods like RNN or LSTM with attention mechanisms for improved sequential crop yield prediction. They also suggest applying transfer learning using pre-trained models to optimize resource usage in similar yield prediction tasks. Shahhosseini et al. [24] combined crop modeling and ML to improve corn yield prediction in the US Corn Belt. ML models with APSIM variables reduced yield forecast RMSE by 7–20%, emphasizing the importance of APSIM factors like soil moisture for accurate predictions and suggesting additional soil water-related variables for enhanced accuracy in the central US Corn Belt.

Huang et al. [25] introduced a dual-stream deep-learning neural network for China's winter wheat yield prediction, integrating remote sensing, weather, and soil data. The model outperformed traditional ML methods with an R2 of 0.79 and 650.21 kg/ha error. With a 13% error rate two months before harvest, it reliably forecasted in-season yields. This multi-source approach offers a valuable tool for large-scale county-level winter wheat prediction. Batool et al. [26] compared AquaCrop simulation and ML techniques for tea yield prediction in Pakistan. XGBoost had MAE 0.123 t/ha, MSE 0.024 t/ha, RMSE 0.154 t/ha. ML outperformed AquaCrop in yield prediction with less data, offering an AquaCrop-ML blend for enhanced tea yield forecasting. Further research should consider all model variables and detailed field data for better outcomes. The comparative analysis of existing literature is given in Table 1.

The review of various methodologies for crop yield prediction underscores the continuous evolution and refinement of predictive models in agricultural research. While traditional models, as discussed previously offer a foundational approach, their accuracy is often limited by the availability and proportionality of the input data, highlighting the necessity for more comprehensive datasets that include high-dimensional genotype data, plant traits, and satellite imagery. The proposed model, which integrates advanced feature extraction, correlation, and ranking techniques with an deep integerated model composed of Bi-GRU and time series CNN, demonstrates superiority in several aspects. With the integration of these model the proposed model provides a highly accurate and robust framework for crop yield prediction, addressing the limitations of previous methodologies and paving the way for more reliable agricultural planning and management.

**Table 1. Comparative analysis of existing literature.**

| Ref. | Methodology | Dataset | Accuracy | Limitations |
|---|---|---|---|---|
| [20] | Regression Modal | Custom dataset from multiple countries | RMSE 8% | • Limited publicly available information on genotype |
| [21] | hybrid fuzzy neural network and deep belief network | Indian Meteorological department, | 92% | • Incorporating variations in hyperparameter assignments, pest infestations, and crop damage into the existing framework can help build a more robust model. |
| [22] | Genetic algorithm (GA)assisted neuro-evolution approach | The Syngenta Crop Challenge dataset | RMSE by 4% and 5%, respectively | • Perturbation on the evolution of the neural network can improve results. |
| [23] | RF,SVM, Gradient Decent, long short term memory, lasso regression | Rajasthan Government | 96% With 0.035 RMSE | • deep learning models on larger datasets is necessary, and integrating remote sensing data with district-level statistical data can enhance the results. |
| [24] | Deep Learning with LSTM | Indian agricultural website | LSTM with 86.3% | • The system can be improved by implementing it on more data, hosting the web application on Google Cloud, and storing data in cloud buckets. |
| [25] | Machine Learning with LSTM | Agricultural Statistical Yearbook, | R2 ranging from 0.77 to 0.87 | • The model can be further enhanced by integrating crop models, incorporating more detailed farming management data. |

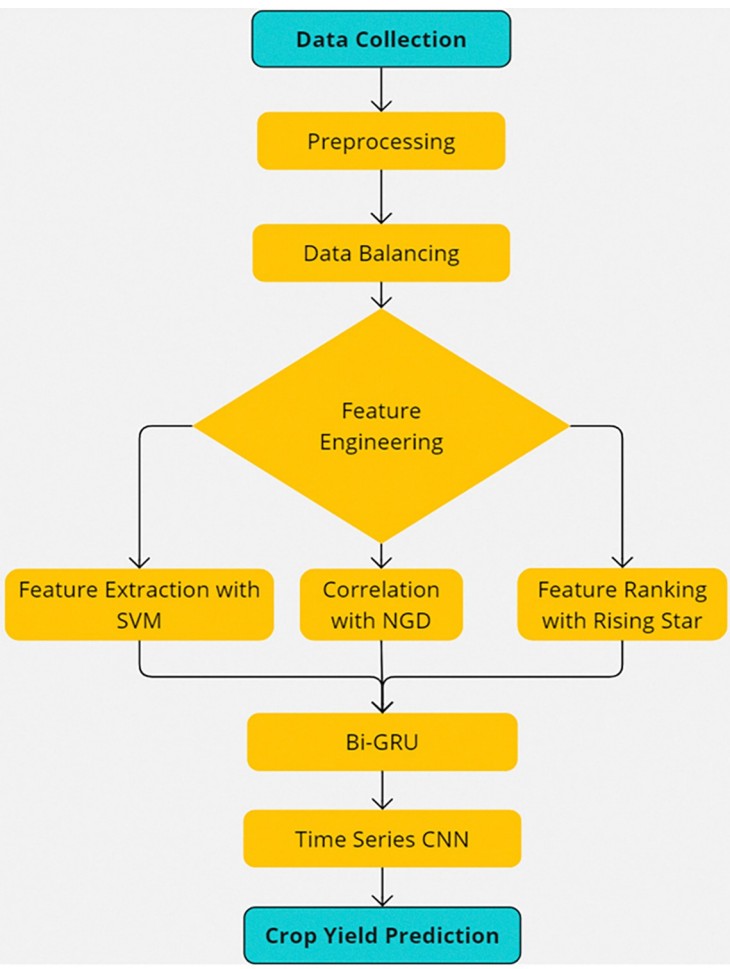

**Fig 2. The working flow of ECP-IEM.**

## 3. Proposed method

The section presented the core methodology of proposed used for this research is presented in this section. We describe the procedures to forecast the yield of crops. Overall idea is presented in Fig 2.

### 3.1 Data collection

Three different dataset that are (ECP-DSI, ECP-DSII and ECP-DSIII) have been used in this study. The graphical distribution of each dataset has been presented in Fig 3. Where the ECP-DSI dataset collected by Khaki, Wang, and Archontoulis [27]. The research region encompasses soybean cultivation across nine states in the United States. This dataset comprises meteorological conditions, soil characteristics, and agricultural practices. It spans from 1980 to 2018, detailing average crop yields throughout this period.

The ECP-DSII dataset consists of the 10 most consumed crops yield in the world. Data is collected from 1990 to 2016. The data is collected from FAO (Food and Agriculture Organization) and World Bank. Whereas, the ECP-DSIII includes information about crop yields, harvested areas, and production quantities for wheat, maize, rice, and soybeans. Agricultural output is measured as production per unit of harvested land area. Often, yield information is

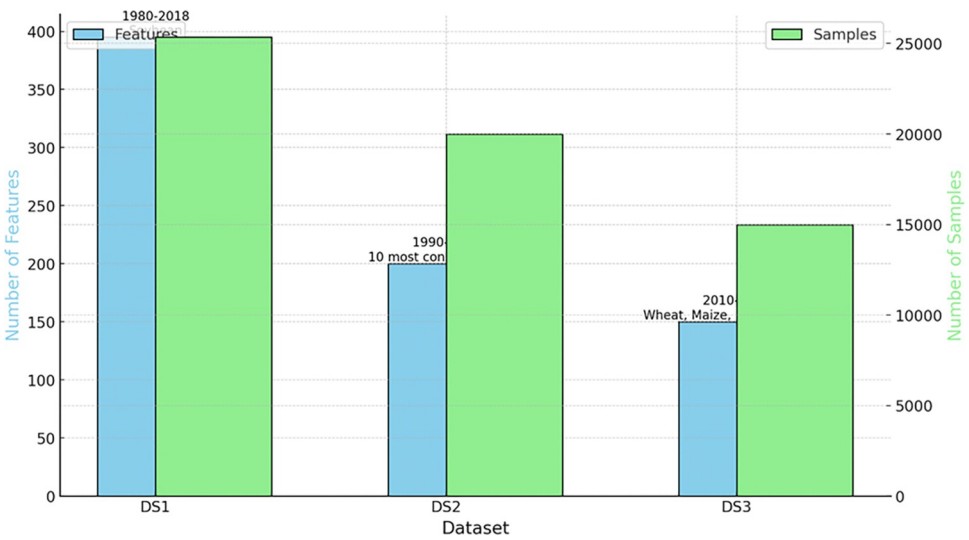

**Fig 3. The overview of ECP-DSI, ECP-DSII and ECP-DSIII.**

derived by dividing total production by the area harvested. This metric is provided for wheat, maize, rice, and soybeans, with yields quantified in tons per hectare. The dataset includes production information spanning from 2020 to 2024.

## 3.2 Preliminary pre-processing

Preprocessing of ECP-IEM involves several steps to ensure that the data is of high quality and suitable for accurate crop yield improvement. Initially, the raw data from various sources, including weather, soil, and management datasets, undergoes thorough cleaning to address missing values, outliers, and inconsistencies. Missing values are imputed using interpolation techniques or statistical methods, while outliers are detected and treated using appropriate statistical measures. This ensures that the data is complete and reliable.

Following data cleaning, normalization or standardization is applied to scale the features, ensuring that all data points are on a comparable scale. This step is vital because features measured on different scales can adversely affect the performance of machine learning models. For normalization, the min-max normalization formula $\frac{X-X_{min}}{X_{max}-X_{min}}$. Where X represents the original value, and $X_{min}$ and $X_{max}$ denote the minimum and maximum values of the feature, respectively. This transformation scales the feature values to a range between 0 and 1, facilitating better model performance. Encoding categorical variables is another essential aspect of preprocessing. Categorical data, such as soil types or crop varieties, is transformed into a numerical format using techniques like one-hot encoding. This process converts categorical values into binary vectors, allowing the model to process and learn from these features effectively. Additionally, temporal alignment is performed to synchronize data points from different sources, ensuring consistency over time. This involves aligning weather data, soil data, and yield records to the same temporal scale, which is critical for capturing the temporal dependencies in crop growth.

## 3.3 Data balancing

After preprocessing, data balancing has been performed to addresses any imbalances in the dataset, such as an unequal distribution of classes in a classification problem [28]. In this work

Adaptive Synthetic Sampling (ADASYN) [29] has been applied that is an effective technique designed to address class imbalances by generating synthetic samples for the minority class in a data-driven manner. This approach is particularly useful for our crop yield prediction model, where the minority class (e.g., years or regions with significantly lower yields) is underrepresented.

After the preprocessing that scales the feature values to a range between 0 and 1, facilitating better model performance. Encoding categorical variables is another essential aspect of preprocessing. Categorical data, such as soil types or crop varieties, is transformed into a numerical format using techniques like one-hot encoding. This process converts categorical values into binary vectors, allowing the model to process and learn from these features effectively. Additionally, temporal alignment is performed to synchronize data points from different sources, ensuring consistency over time. This involves aligning weather data, soil data, and yield records to the same temporal scale, which is critical for capturing the temporal dependencies in crop growth. Dimensionality reduction is also a key component of the preprocessing pipeline. By using feature selection process which is described in next section has been applied to remove redundant and irrelevant features, retaining only the most informative variables. This step reduces the computational complexity of the model and enhances its ability to generalize and learn meaningful patterns from the data.

Let $D = \{(x_i, y_i)\}_{i=1}^{N}\}^{i=1i=1}$ be the dataset, where $x_i$ represents the feature vector (including weather, soil, and management features), and $y_i \epsilon \{0,1\}$ denotes the class label, with 1 being the minority class (e.g., low yield) and 0 being the majority class (e.g., high yield). Define $N_{min}$ and $N_{maj}$ as the number of samples in the minority and majority classes, respectively. To calculate the imbalance ratio $r$ as depicted in Eq 1:

$$r = \frac{N_{maj}}{N_{min}} \qquad (1)$$

For each minority class sample $x_i$, find its k-nearest neighbors using Euclidean distance [30] for continuous feature. Let $d_i$ be the number of majority class samples among these k-nearest neighbors. Determine the sampling distribution for generating synthetic samples, the ratio $r_i$ for each minority class sample $x_i$ could be computed as depicted in Eq 2:

$$r_i = \frac{d_i}{k} \qquad (2)$$

The number of synthetic samples $G_i$ to be generated for each minority sample $x_i$ is given by: $G_i = r_i \times G$ where G is the total number of synthetic samples needed, typically set as $N_{maj} - N_{min}$. For each minority class sample $x_i$, generate $G_i$ synthetic samples. Select a random neighbor $x_{i,nn}$ from the k-nearest neighbors and interpolate to create a synthetic sample $x_{new}$ depicted in Eq 3:

$$x_{new} = x_i + \delta \times (x_{i,nn} - x_i) \qquad (3)$$

where $\delta$ is a random number drawn from the uniform distribution $\delta \sim U(0,1)$.

Integrate the generated synthetic samples with the original dataset to form a balanced training set. Train machine learning models on this balanced dataset to improve the detection of heart diseases. Finally, train machine learning models on this balanced dataset to improve crop yield predictions. The models benefit from the enriched dataset, which now has a more balanced representation of different yield levels, thereby enhancing their ability to generalize and make accurate predictions. The improved dataset allows the models to learn from a wider variety of examples, leading to better performance in predicting crop yields across different

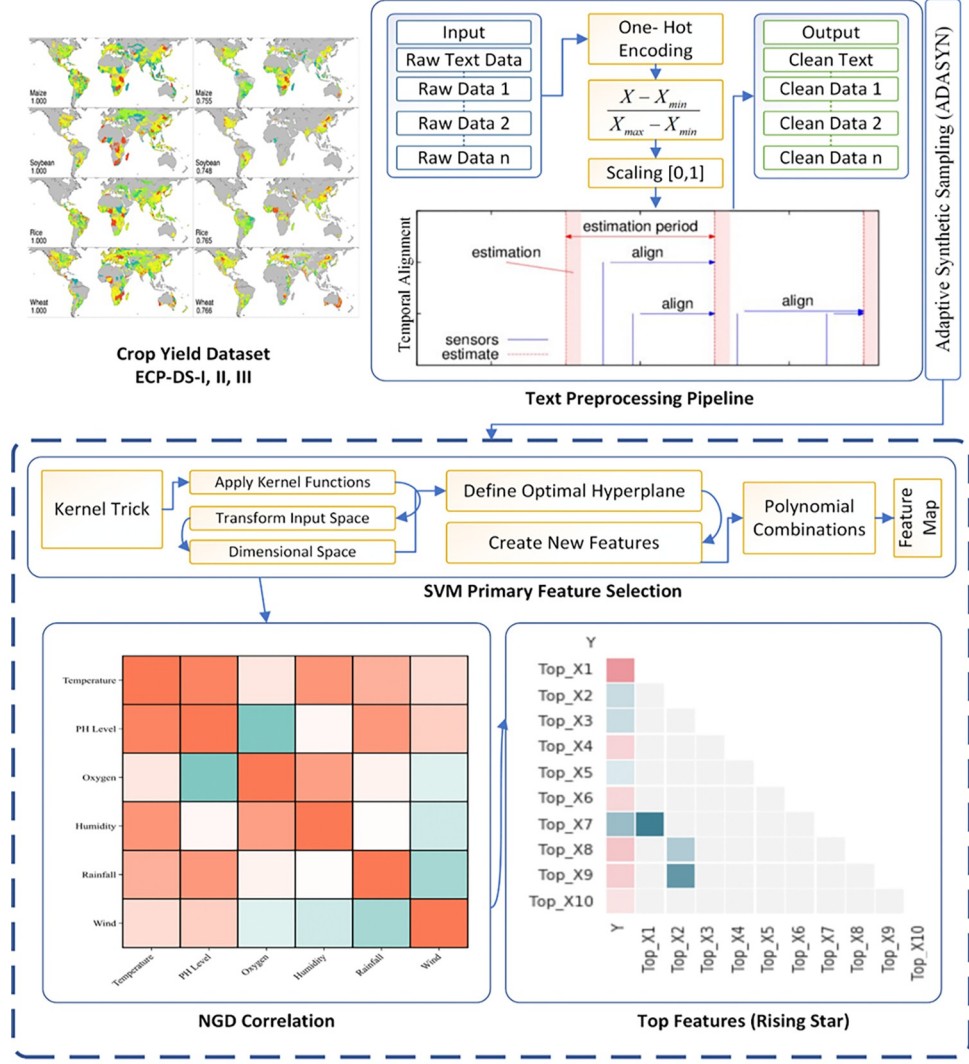

**Fig 4. The ECP-IEM process from data input to feature engineering.**

scenarios and conditions. This approach ultimately contributes to more reliable and robust crop yield predictions, aiding in better agricultural planning and resource management. The ECP-IEM process from data input to feature engineering is shown in Fig 4.

## 3.4 Feature engineering

The feature engineering process ECP-IEM as depicted in Algorithm 1, involves a comprehensive approach to selecting, extracting, and refining the most relevant features from the dataset. This step is essential for enhancing the model's predictive accuracy and robustness. By leveraging advanced techniques such as SVM for initial feature selection, NGD for correlation analysis, and the Rising Star algorithm for feature ranking, ECP-IEM ensure that the final set of features is both highly informative and non-redundant. This systematic approach to feature engineering allows the model to capture the complex interactions and dependencies within the data, ultimately leading to more accurate and reliable crop yield predictions.

**Algorithm 1:** Feature Engineering Including Feature Extraction, Corre-
lation and Ranking

**Input:**

• *Dataset* $D = \{(x_i, y_i)\}_{i=1}^{N})\}^{i=1\,i=1}$ *such that* $x_i$ *and* $y_i$ *∈Class Label*

• *Parameters for SVM, NGD, and Rising Star algorithm*: *k (number of nearest neighbors)*, *α, β*.

  **Output:** *$f_i$ as Selected and ranked features.*

1 **Train** SVM on D and find W that weight vector using Optimization

$$W \leftarrow min\frac{1}{2}\|W\|^2 + C\sum_{1=1}^{N}\max(0, 1 - y_i(w \cdot x_i + b))$$

2 **For each** feature j

3   **Do**

Importance(j)←|$W_j$|

4 **End for**

5 Select top m features

6   *SelectedFeatures←{j|Importance(j) is among the top m features}*

7 **Initialize** correlation matrix *C*:

8 **do**

9   *$C_{ij}$ ← 0 for all i,j*

10 **For each** pair of selected feature ($x_i$, $x_j$)

11   **Do**

$$NGD\left(x_i, x_j\right) \leftarrow \frac{max\{log(f(x_i)), log(f(x_j))\} - log(f(x_i \cap x_j))}{log(N) - min\{log(f(x_i)), log(f(x_j))\}}$$

$C_{ij}$←*NGD($x_i$, $x_j$)*

12 **End for**

13 **For each** feature *i*

14   **Do**

$P_i \leftarrow \frac{1}{T}\sum_{t=1}^{T} \text{Weight}_t(i)$

$R_i$←*$Weight_{recent}(i)$*

*Rising Star Score $S_i \leftarrow \alpha P_i + \beta R_i$*

 *RankedFeatures ← sort($\{(i, S_i)\}$) in descending order of $S_i$*

15 **End for**

16 **Return**

The complete procedure of feature extraction for ECP-IEM begins with the use of SVM to identify and select the most relevant features. SVMs are primarily used for classification and regression tasks, but they can also be utilized for feature selection by determining the features that contribute most significantly to the decision boundary. In this context, an SVM classifier is trained on the dataset, and the absolute values of the weights of the linear SVM model are used as a measure of feature importance. Mathematically, given a trained linear SVM model $w \cdot x + b = 0$, where www is the weight vector and b is the bias, the importance of the j[th] feature can be expressed as $|w_j|$. Features with higher absolute weight values are considered more important and are selected for further analysis. This process helps in reducing the dimensionality of the dataset, retaining only those features that have a significant impact on the model's predictions.

In the context of feature selection, the SVM is trained to classify the crop yield data into different categories based on the available features. The resulting weight vector www represents the contribution of each feature to the decision boundary. By examining the absolute values of these weights, we can identify which features have the most influence on the classification outcome. This method ensures that only the most relevant features are retained for further processing, improving the efficiency and performance of the subsequent analysis steps. Following feature extraction using SVM, the next step involves evaluating the correlation between the selected features using NGD. NGD is a semantic similarity measure that quantifies the relatedness of concepts based on their co-occurrence in search engine results. For any two features $C_i$

and $C_j$, NGD is calculated as depicted in Eq 4:

$$NGD\left(C_i, C_j\right) = \frac{\max\{\log(f(C_i)), \log(f(C_j))\} - \log(f(C_i \cap C_j))}{\log(N) - \min\{\log(f(C_i)), \log(f(C_j))\}} \tag{4}$$

where $f(C_i)$ and $f(C_j)$ are the frequencies of $C_i$ and $C_j$ appearing in search results, $f(C_i, C_j)$ is the frequency of their co-occurrence, and NNN is the total number of web pages indexed by the search engine. Lower NGD values indicate higher similarity between features. By calculating NGD for all pairs of selected features, a correlation matrix is constructed, helping to understand the relationships and dependencies among the features. This step is crucial because it allows us to identify and potentially eliminate redundant features that do not provide additional information beyond what is already captured by other features. Fig 5 shows some of the obtained feature visualization that has been generated based on the working of algorithm 1.

The NGD-based correlation analysis provides a deeper understanding of how the selected features interact with each other. For instance, if two features have a high semantic similarity (i.e., a low NGD value), they may convey similar information about the crop yield. In such cases, retaining both features might be unnecessary, and one of them can be discarded to reduce redundancy. On the other hand, features with low semantic similarity (i.e., high NGD values) are likely to provide unique and complementary information, making them valuable for the model. This correlation analysis ensures that the final set of features is both comprehensive and non-redundant, enhancing the model's ability to capture diverse aspects of the data. The final step involves ranking the features using the Rising Star algorithm, which assesses the significance of features based on their historical performance and recent trends. The Rising Star algorithm assigns a score to each feature, considering both its past importance and its current relevance. The score $S_i$ for feature $i$ is calculated as depicted in Eq 5:

$$S_i = \alpha \cdot P_i + \beta \cdot R_i \tag{5}$$

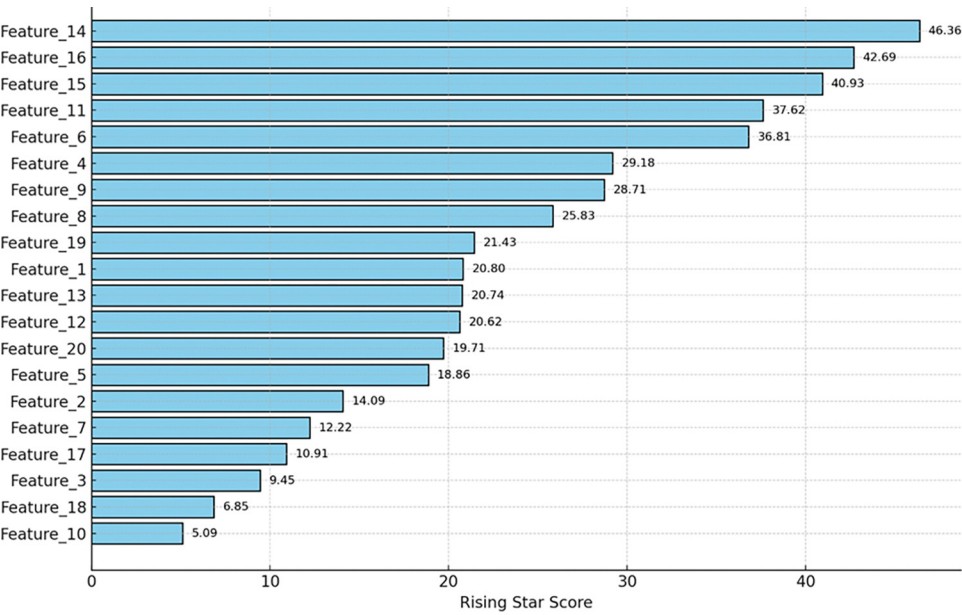

**Fig 5. Feature engineering having ranked feature based on rising star score.**

Where $P_i$ represents the historical performance of the feature, $R_i$ denotes its recent performance, and $\alpha$ and $\beta$ are weighting factors that balance the contribution of historical and recent performance. The historical performance $P_i$ can be measured by the average weight of the feature in previous models, while the recent performance $R_i$ can be derived from its weight in the most recent model. Features with higher scores are ranked higher, indicating their greater importance in the prediction model.

The Rising Star algorithm integrates both long-term and short-term performance metrics to provide a balanced evaluation of each feature's importance. Historical performance $P_i$ reflects the feature's consistent contribution to model accuracy over time, while recent performance $R_i$ captures the feature's current relevance based on recent data and model iterations. The weighting factors $\alpha$ and $\beta$ can be adjusted to emphasize either historical stability or recent trends, depending on the specific requirements of the model and the nature of the data.

By integrating SVM-based feature selection, NGD-based correlation analysis, and Rising Star-based feature ranking, the procedure ensures that the most relevant and impactful features are identified and prioritized for inclusion in the model. This comprehensive approach enhances the model's ability to capture meaningful patterns and dependencies, leading to more accurate and robust crop yield predictions. The combination of these advanced techniques allows the model to effectively handle the complexity and variability of agricultural data, ultimately contributing to better decision-making and resource management in crop production. The model of ECP-IEM is shown in Fig 6.

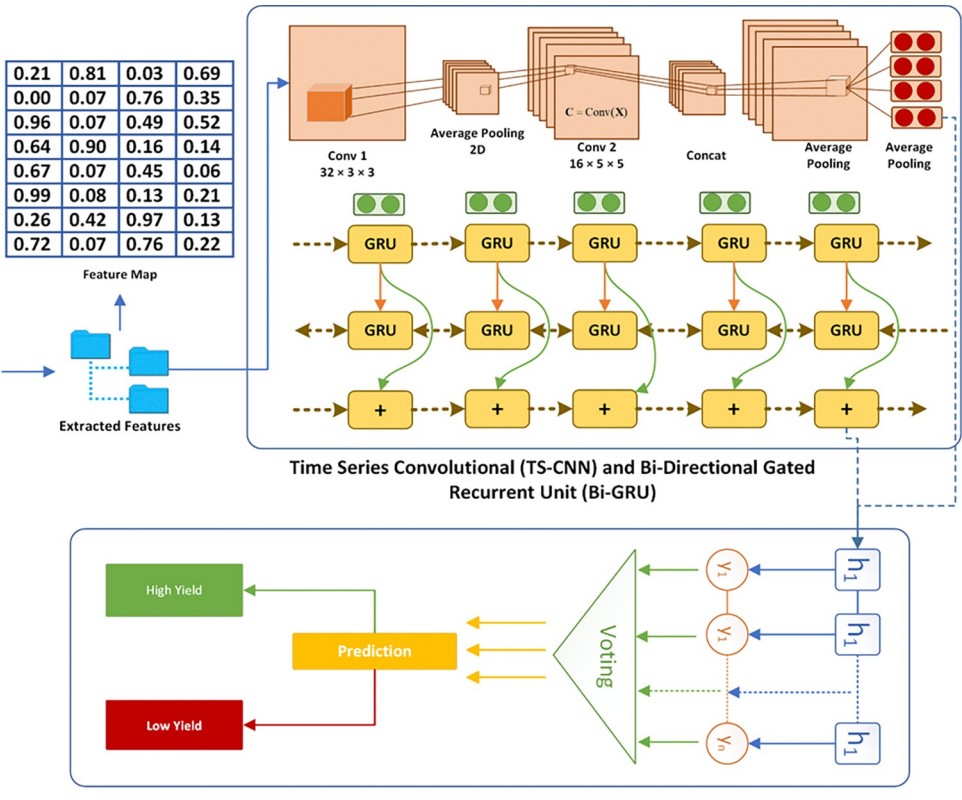

**Fig 6. ECP-IEM based crop yield prediction.**

## 3.5 Prediction via deep integrated classifier

The final prediction in the proposed model is generated using a deep integrated classifier that integrates the strengths of Bi-GRU and time series and CNN. This approach leverages different aspects of the data, capturing temporal, spatial, and enhanced feature representations to produce a robust and accurate crop yield prediction.

The Bi-GRU model processes sequential data such as time-series weather and soil conditions. Given an input sequence $X = [x_1, x_2, \ldots, x_T]$. where $x_t$ represents the feature vector at time t, the Bi-GRU captures temporal patterns and dependencies. The forward and backward GRUs process the sequence to generate hidden states $\overleftarrow{h_t}$ and $\overrightarrow{h_t}$, respectively. The final hidden state $h_T$ is obtained by concatenating the forward and backward hidden states depicted in Eqs (6–7):

$$h_t = B_i GRU(x_i, h_{t-1}) \tag{6}$$

$$h_T = \left[ \overleftarrow{h_t}, \overrightarrow{h_t} \right] \tag{7}$$

In Eqs (8–11), the BI-GRU calculation is shown by denoting the sigmoid operation, hidden states & input vectors as $ft^*$, $hi_t$ and $v_t$ respectively. Reset data is denoted by $r_t$, WG reveals the weight factor, $t$ reveal the time interval.

$$F_t = ft^*(WG_i * [hi_{t-1}, v_t]) \tag{8}$$

$$r_t = ft^*(WG_i * [hi_{t-1}, v_t]) \tag{9}$$

$$hi_t = \tanh(WG_c * [r_t hi_{t-1}, v_t]) \tag{10}$$

$$hi_t = (1 - F_t).C_{t-1} + F_t hi_t \tag{11}$$

This output represents the temporal features extracted from the input sequence.

The CNN model focuses on extracting spatial features from data such as satellite images or spatial patterns in weather data. The input to the CNN is a matrix $I$ representing the spatial data. The CNN applies multiple layers of convolutions, pooling, and activation functions to extract high-level spatial features. Each convolutional layer applies a set of filters to the input data, detecting various spatial patterns, and pooling layers reduce the dimensionality while preserving the most important features. The CNN model is described as per Eq (12) which gets the $ft^*$ as input. In Eq (13), $\pi_w^l \rightarrow$ weight that is optimally tuned, $B_w^l \rightarrow$ bias of $w^{th}$ flter linked to $i^{th}$ layer. The activation value ($act_{r,t,w}^l$) related to convolutional features $D_{r,t,w}^l$ is shown by Eq (13).

$$D_{r,t,w}^l = \pi_w^{lT} ft^* + B_w^l \tag{12}$$

$$act_{r,t,w}^l = act(D_{r,t,w}^l) \tag{13}$$

Eq (14) shows $H_{r,t,w}^l$ calculation. CNN loss PL is revealed by Eq (15), in which, θ signifies

the term associated with $W_w^l$ and $B_w^l$.

$$H_{r,t,w}^l = pool(act_{m,h,w}^l), \forall (m,h) \in nn_{r,t} \tag{14}$$

$$PL = \frac{1}{wn} \sum\nolimits_{h=1}^{wn} l\left(\theta; H^{(h)}, F^{(h)}\right) \tag{15}$$

The final feature map F captures the spatial information relevant to crop yield.

After receiving results from each model, every model vote for its predicted class label. The class with the most votes becomes the final prediction. Let us assume that the forecast of every individual model is labeled as: $y_{CNN}$ and $y_{Bi-GRU}$ respectively. The last prediction after voting will be called $y_{final}$.

Each model votes for the predicted class using Eq 16.

$$y_{final} = \arg max_c(\Sigma_{i=1}^n 1_{ycnn[i]=c+} \Sigma_{i=1}^n = c) \tag{16}$$

For example, if two models predict a high yield (class 1) and one model predicts a low yield (class 0), the final prediction will be a high yield (class 1) because it has the majority of votes. This approach ensures that the final prediction leverages the strengths of each individual model, providing a robust and accurate forecast for crop yield. This methodology, integrating advanced machine learning and deep learning techniques, aligns with your expertise in designing models to improve seasonal crop yield, thus offering a practical and effective solution for agricultural management.

## 3.6 Final recommendations

Incorporating the insights derived from the advanced predictive model, farmers and agricultural stakeholders can implement a range of strategic recommendations to enhance crop yield. These recommendations are designed to optimize various aspects of crop management, including planting schedules, irrigation practices, fertilization, pest and disease control, crop variety selection, resource allocation, and real-time monitoring. By leveraging the model's accurate predictions and detailed analyses, the following strategies can maximize productivity, ensure efficient resource use, and mitigate potential risks, ultimately leading to more sustainable and profitable agricultural practices.

- Utilize the predictions to determine the best planting times based on weather forecasts and soil conditions. Adjust planting schedules to ensure that crops are sown at the most favorable times to maximize germination and early growth.

- Use the model's insights to implement precise irrigation schedules, ensuring that crops receive adequate water at critical growth stages. This prevents both water stress and over-irrigation, which can lead to reduced yields.

- Apply fertilizers based on the predicted nutrient requirements at different stages of crop growth. This ensures that crops receive the necessary nutrients without over-fertilization, which can harm the environment.

- Implement proactive pest and disease control measures by leveraging the model's ability to predict potential outbreaks. Early intervention can significantly reduce crop damage and improve yields.

- Use the predictions to choose crop varieties that are best suited to the expected environmental conditions. Selecting resilient and high-yield varieties can enhance overall productivity.

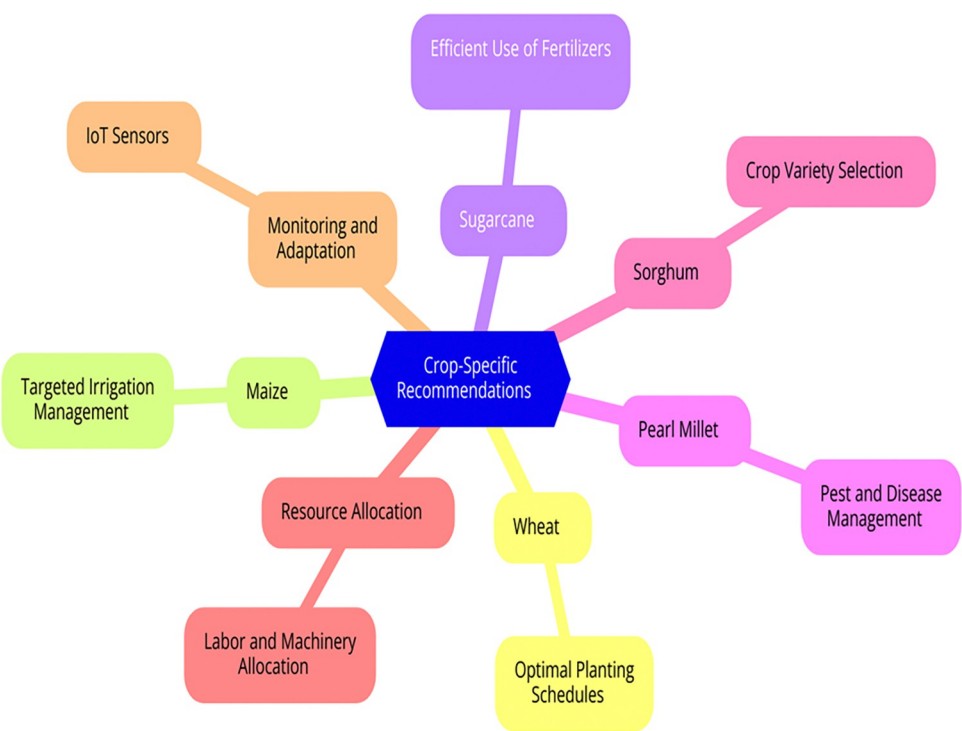

**Fig 7. Seasonal crop yield improvement based on ECP-IEM Recommendation.**

- Optimize the allocation of resources such as labor, machinery, and capital based on the predicted workload and yield. This ensures that resources are used efficiently and effectively.

- Continuously monitor crop conditions using IoT sensors and remote sensing technologies. Adapt farming practices in real-time based on the latest predictions to address any emerging issues promptly.

Fig 7 shows the Incorporating insights derived from the advanced predictive model, farmers and agricultural stakeholders can implement a range of strategic recommendations to enhance crop yield. These recommendations are designed to optimize various aspects of crop management, including planting schedules, irrigation practices, fertilization, pest and disease control, crop variety selection, resource allocation, and real-time monitoring. By leveraging the model's accurate predictions and detailed analyses, these strategies aim to maximize productivity, ensure efficient resource use, and mitigate potential risks, ultimately leading to more sustainable and profitable agricultural practices.

## 4. Experimental evaluation

This section provides details of the experiments and their results. The dataset used, the performance metrics and baseline approaches are presented.

The dataset was first split into training and testing sets before the application of any data balancing techniques. To address class imbalance, only the training set was balanced using the Adaptive Synthetic Sampling (ADASYN) method, while the testing set remained untouched to serve as an uncontaminated evaluation set. This approach ensures the integrity of the testing set, allowing for an unbiased assessment of the model's performance. Additionally, k-fold cross-validation was employed to enhance the robustness of the model evaluation. Within

each fold, the ADASYN technique was applied exclusively to the training portion, while the validation fold remained unaltered. This strategy maintains the purity of the validation data, preventing any information leakage from the training set. By strictly ensuring that no information from the testing or validation sets influences the training process, this methodology provides a more realistic and reliable reflection of the model's generalization capability in real-world scenarios.

## 4.1 Datasets

The datasets utilized in this research are described in Table 2, analyzing and predicting crop yields within the Corn Belt region of the United States, as well as globally for key crops. The first dataset includes observed average yield data for corn and soybeans from 1980 to 2018, spanning 1,176 counties for corn and 1,115 counties for soybeans. These states predominantly cultivate corn and soybeans. This dataset provides valuable historical yield performance insights, crucial for understanding trends and variations in crop yields over time. Crop yield prediction is a significant agricultural challenge influenced by various factors, including weather conditions (rainfall, temperature), pesticide usage, and historical crop yield data. The second dataset, sourced from the Food and Agriculture Organization (FAO) and the World Data Bank, includes comprehensive weather data, pesticide usage, and historical yield information. Integrating this data is essential for agricultural risk management and making accurate future yield predictions.

The third dataset provides data on crop yields, harvested areas, and production quantities for wheat, maize, rice, and soybeans. This dataset offers a global perspective on crop production, measured in tons per hectare, for the four major crops: wheat, maize, rice, and soybeans. These datasets provide a robust foundation for analyzing crop yields, understanding historical trends, and developing predictive models. The integration of regional and global data ensures comprehensive coverage and facilitates the assessment of various factors influencing crop yields.

## 4.2 Performance matrices

To evaluate the performance of MDD the following benchmark matrices has been used.

- **Accuracy:** Accuracy is the ratio of correctly predicted instances to the total instances. It is a measure of the overall effectiveness of a classification model.

$$Accuracy = \frac{True\ Positive + True\ Negative}{Total\ Sample} \tag{17}$$

- **Precision:** Precision, or Positive Predictive Value, is the measure of correctly identified positive observations in relation to all observations predicted as positive. It reflects the

**Table 2. Dataset description.**

| Dataset | Crops | URLs |
|---|---|---|
| ECP-DS-I | Corn and Soybeans | https://widgets.figshare.com/articles/27612663/embed?show_title=1 |
| ECP-DS-II | Different Crops | https://www.kaggle.com/datasets/patelris/crop-yield-prediction-dataset |
| ECP-DS-III | wheat, maize, rice, and soybeans. | https://www.kaggle.com/datasets/thedevastator/the-relationship-between-crop-production-and-cli |

proportion of true positive cases among the predicted positive instances, indicating the accuracy of positive predictions.

$$Precision = \frac{True\ Positive}{True\ Positive + False\ Positive} \tag{18}$$

- **Recall:** It may also know as Sensitivity or True Positive Rate, is the ratio of correctly predicted positive observations to the all observations in the actual class.

$$Recall = \frac{True\ Positive}{True\ Positive + False\ Negative} \tag{19}$$

- **F1 Score:** The F1 Score represents the harmonic mean of Precision and Recall, offering a single metric that balances the trade-off between these two measures. It is particularly valuable when considering both false positives and false negatives, providing a comprehensive assessment of a model's performance.

$$F1\ Score = 2 \times \frac{Precision \times Recall}{Precision + Recall} \tag{20}$$

- **Mean Absolute Error (MAE):** This measure calculates the mean size of errors in a set of predictions, disregarding their direction. It represents the average of the absolute differences between predicted values and actual observations in the test sample, giving equal importance to all individual discrepancies. The formula for MAE is:

$$MAE = \frac{1}{n}\Sigma_{i=1}^{n}|y_i - \hat{y}_i| \tag{21}$$

- **Mean Squared Error (MSE):** This measure calculates the mean of the squared errors. As the second moment of the error about the origin, it accounts for both the variance and bias of the estimator. The formula for MSE is:

$$MSE = \frac{1}{n}\Sigma_{i=1}^{n}\left(y_i - \hat{y}_i\right)^2 \tag{22}$$

- **Root Mean Square Error (RMSE):** This is a common method for evaluating a model's prediction error in quantitative data. It involves taking the square root of the mean of the squared differences between observed and predicted values.

$$RMSE = \sqrt{MSE} \tag{23}$$

## 4.3 Baseline method

To evaluate the performance of the proposed model, the following baseline has been selected due to its proficiency.

- Baseline Approach 1 [31]: Created a tree-based ensemble learning model to predict crop suitability and productivity.

- Baseline Approach 2 [32]: Employed LSTM recurrent neural networks and 1DCNN for crop forecasting.

- Baseline Approach 3 [33]: Introduced a stacking-based ensemble deep learning technique called Model Agnostic Meta-Learning (MAML) for classification purposes.

## 4.4 Results

The proposed model was evaluated on three different datasets (ECP-DS-I, ECP-DS-II, and ECP-DS-III) to assess its performance. The results demonstrated high accuracy, precision, and recall across all datasets, indicating the model's robustness and reliability for crop yield prediction as shown in Fig 8. On ECP-DS-I, the model achieved an accuracy of 95.76%, with a precision of 94.05% and a recall of 94.98%, highlighting its ability to accurately predict crop yield with minimal false positives and false negatives. Performance slightly improved on ECP-DS-II, with the model reaching an accuracy of 96.23%, precision of 94.42%, and recall of 94.12%, demonstrating consistency across different datasets and underscoring the model's generalizability. The highest performance was observed on ECP-DS-III, where the model achieved an accuracy of 97.22%, a precision of 95.34%, and a recall of 96.01%, showcasing its optimal capability to predict crop yield accurately. Overall, the proposed model showed very good results across all datasets, significantly improving compared to baseline models. The high accuracy, precision, and recall values indicate that the model is well-suited for early crop yield prediction, providing valuable insights for farmers to enhance crop production and ensure food security.

To design the architectures for CNN and Bi-GRU, we conducted an iterative process involving hyperparameter tuning and architecture refinement. For the CNN, we tested various configurations, adjusting the number of convolutional layers, kernel sizes, and pooling operations to optimize spatial feature extraction from image data. For the Bi-GRU, we explored different numbers of hidden units, time steps, and learning rates to effectively capture temporal patterns. Experimental comparisons of the individual performances of CNN and Bi-GRU revealed that the CNN achieved an accuracy of 92.13% and Bi-GRU reached 91.45%. However, the combined fusion model improved the accuracy to 96.34%. This fusion leverages the

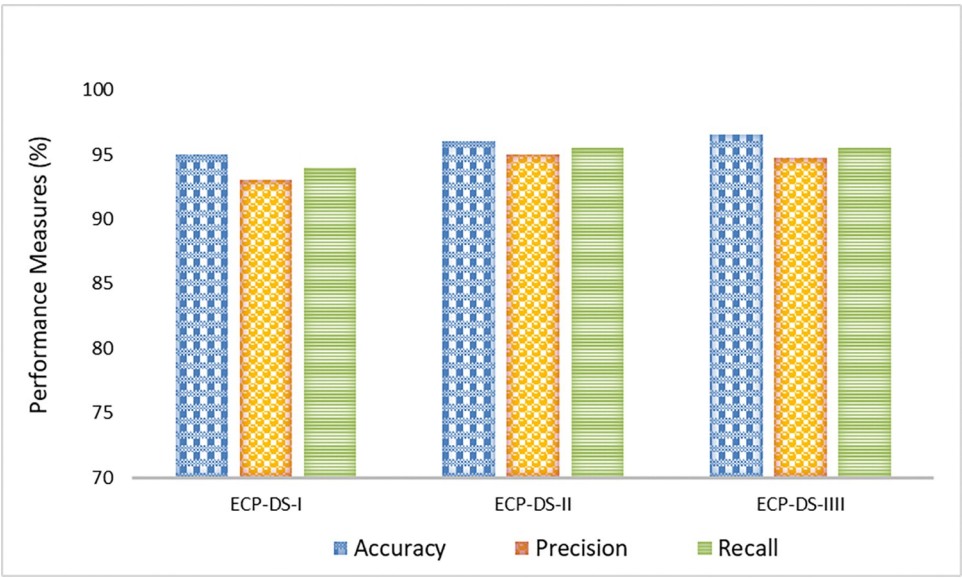

**Fig 8. Experimental results in terms of accuracy, precision and recall.**

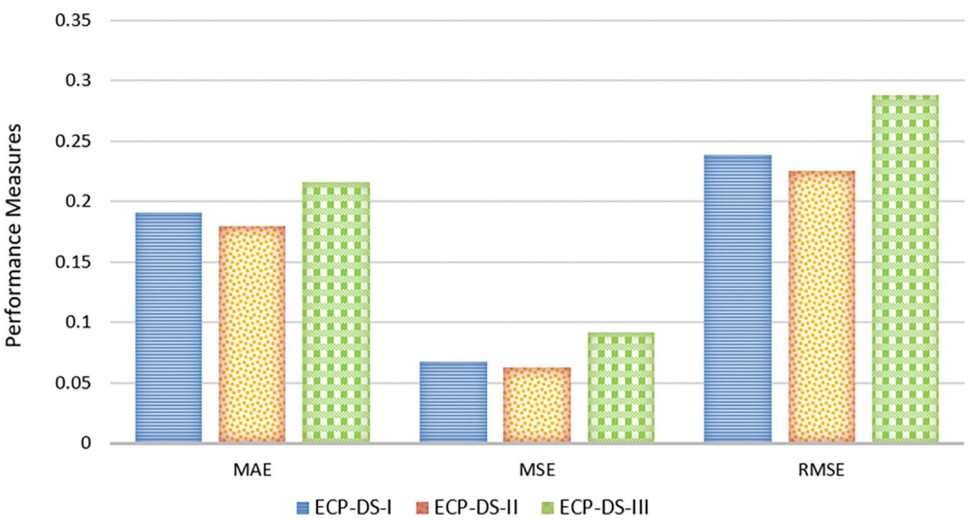

**Fig 9. Performance through MAE, MSE and RMSE.**

strengths of both models, leading to better overall performance, as reflected in improved precision, recall, and other metrics across all datasets. Fig 9 presents the performance of three datasets, ECP-DS-I, ECP-DS-II, and ECP-DS-III, evaluated using Mean Absolute Error (MAE), Mean Squared Error (MSE), and Root Mean Squared Error (RMSE). The results indicate that for MAE, ECP-DS-I has a relatively high value compared to ECP-DS-II but is lower than ECP-DS-III, with ECP-DS-II having the lowest MAE among the three datasets.

Regarding MSE, ECP-DS-I shows the lowest value, followed by ECP-DS-II, with ECP-DS-III having the highest MSE. For RMSE, ECP-DS-II again performs the best with the lowest value, while ECP-DS-I has a higher RMSE than ECP-DS-II but lower than ECP-DS-III, which exhibits the highest RMSE. In summary, ECP-DS-II demonstrates superior performance with the lowest error metrics in both MAE and RMSE, indicating the most accurate predictions overall. ECP-DS-I shows the best performance in MSE, while ECP-DS-III consistently records the highest error metrics across all three measures, suggesting it is the least accurate among the datasets.

The proposed model demonstrates a significant enhancement in crop yield forecasting compared to traditional baseline models, as evidenced by improvements in accuracy, precision, and recall. The accuracy of the proposed model stands at 96.23%, a substantial increase over the baseline models, which range from 91.34% to 93.22%. This 3.01% to 4.89% boost in accuracy highlights the proposed model's superior ability to correctly classify crop yield predictions. Precision also shows marked improvement, with the proposed model achieving 94.42%, surpassing the baseline models' precision rates of 89.78% to 91.34%.

This 3.08% to 4.64% increase underscores the model's enhanced reliability in identifying true positives and reducing false positives. Similarly, the proposed model's recall of 94.12% outperforms the baseline models, which range from 90.23% to 91.89%, representing an increase of 2.23% to 3.89%. This improvement in recall reflects the model's ability to capture a higher proportion of true positives and minimize false negatives. Overall, the proposed model's superior performance across all metrics accuracy, precision, and recall demonstrates its effectiveness in providing more reliable and accurate crop yield forecasts, thereby supporting better decision-making for farmers and contributing to enhanced food security. The comparative analysis is illustrated in Fig 10.

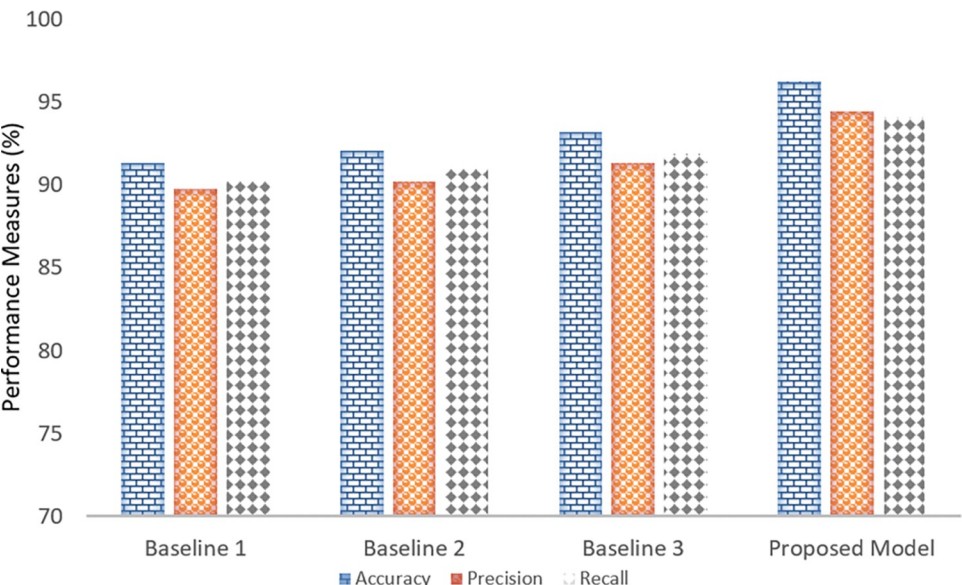

**Fig 10. Comparative analysis of proposed model with baseline approaches in terms of accuracy, precision and recall.**

The proposed model significantly outperforms the baseline models in terms of prediction accuracy, as demonstrated by its performance across Mean Absolute Error (MAE), Mean Squared Error (MSE), and Root Mean Squared Error (RMSE) as shown in Fig 11. The proposed model achieves an MAE of 0.191, which is lower than the baseline models, whose MAE values range from 0.199 to 0.314. This reduction indicates that the proposed model provides more accurate average predictions for crop yield. In terms of MSE, the proposed model reports the lowest value at 0.0674, surpassing the baseline models that range from 0.071 to 0.172. This decrease in MSE reflects the model's superior ability to reduce the squared deviations between

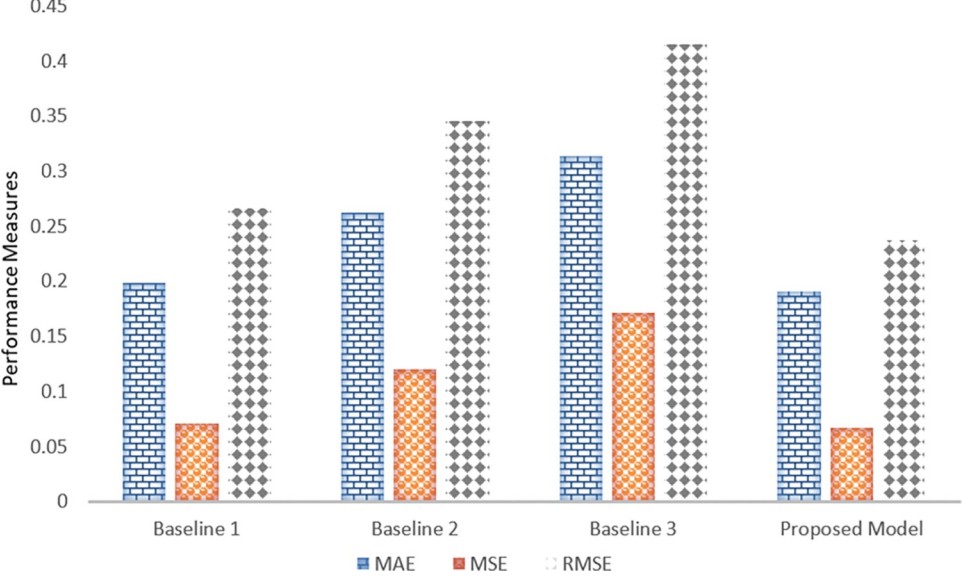

**Fig 11. Comparative analysis of proposed model with baseline approaches in terms of MAE, MSE and RMSE.**

predicted and actual values. The proposed model also excels in RMSE, with a value of 0.238, compared to the baseline models' RMSE values that span from 0.266 to 0.415. The lower RMSE demonstrates the proposed model's effectiveness in minimizing overall prediction errors and deviations. Overall, the proposed model's consistent improvements in MAE, MSE, and RMSE highlight its superior accuracy and reliability in crop yield forecasting, crucial for enhancing agricultural practices and food security.

## 5. Conclusion and future work

Accurate crop yield forecasting is vital for ensuring food security and making informed decisions. With the increasing population and the impacts of global warming, addressing food security has become a priority, making precise yield forecasting critically important. AI has significantly improved yield accuracy. However, existing ML methods often rely on statistical measures such as regression, correlation, and chi-square tests for predicting crop yield. These models tend to show low accuracy when the number of influencing factors such as weather, soil conditions, wind, fertilizer quantity, seed quality, and climate increase. The proposed methodology enhances crop yield prediction through various stages: Data Collection, Preprocessing, Feature Extraction using SVM, Correlation using NGD, and Feature Ranking with the Rising Star algorithm. This study integrates Bi-GRU and time series CNN models to predict crop yield effectively. The proposed model demonstrated superior performance across all datasets, significantly improving upon baseline models. The results indicate the model's robustness and accuracy in forecasting crop yields, providing farmers with early insights into expected yields, thus aiding in strategic planning and resource optimization to enhance crop productivity. Future work should focus on further refining the model by incorporating additional environmental and genetic factors, exploring more advanced deep learning architectures, and integrating real-time data streams for dynamic prediction updates. Additionally, expanding the model's applicability to a wider range of crops and geographic regions will enhance its utility and effectiveness in diverse agricultural settings. By continuing to advance this research, we can contribute to more resilient and sustainable agricultural practices, ultimately supporting global food security efforts.

## Author Contributions

**Conceptualization:** Ghulam Mustafa, Muhammad Ali Moazzam.

**Data curation:** Muhammad Ali Moazzam.

**Formal analysis:** Asif Nawaz.

**Investigation:** Asif Nawaz, Deema Mohammed Alsekait.

**Methodology:** Diaa Salama AbdElminaam.

**Validation:** Deema Mohammed Alsekait.

**Visualization:** Tariq Ali.

**Writing – original draft:** Ghulam Mustafa.

**Writing – review & editing:** Ahmed Saleh Alattas.

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
