## [Decision Letter · Decision Letter 0]

29 Sep 2024

PONE-D-24-34725ECP-IEM: Enhancing Seasonal Crop Productivity with Deep Integrated Ensemble ModelPLOS ONE

Dear Dr. Mustafa,

Thank you for submitting your manuscript to PLOS ONE. After careful consideration, we feel that it has merit but does not fully meet PLOS ONE’s publication criteria as it currently stands. Therefore, we invite you to submit a revised version of the manuscript that addresses the points raised during the review process.

 The novelty of the research is incremental. The authors have combined existing approaches, without clear justification for the choice of the underlying algorithms, and proposed the use of an ensemble approach, i.e., CNN and Bi-GRU. The concern here is a) why these approaches were selected, b) what were the various other approaches that were considered, c) what was the methodology, and experimental results to support the choice, pursued to select the two methods and d) using two underlying systems, one for each modality, does not constitute an ensemble.The research contributions of the work are not related to the novelty and performance, but rather they relate to the impact of the application area, which is indeed common with all other works in this field.The authors should provide a detailed description of the datasets, and the specific challenges associated with each dataset. This will allow them to justify the choice of architectures/methods.The ensemble approach is typically based on using simpler models, which provide for complementary features and decisions, and is characterized by a larger number of models. The larger number of models offers robustness and complementarity demonstrating better performance than that of the underlying models. In this work, there is fusion of two systems, rather than an ensemble, however, there is no substantial evidence (in terms of results) that this leads to better performance than each of the two systems. The recommendation is to consider multiple approaches to extract information from the original image and time series data, which can be combined together into a robust group of features.It is not clear what the specific architectures for the CNN and the Bi-GRU are. What was the process for designing these architectures? The authors should provide information on the design process, including experimental results to support their final architectural recommendations. Ideally, the individual network design performances should be reported.Based on the description of the work, the dataset has been balanced prior to training and testing. This introduces data leakage where information from samples that should be used in validation/testing only, has been used in training, which impacts on the accuracy of the results and indeed, it's generalization. Thus, the experiments need to be repeated and depending on the training strategy, the dataset should be split into training and testing, followed by balancing the training set only, and testing on the uncontaminated testing set or using k-fold cross-validation, where at each instance, only the folds of the training set are balanced, while the fold of the validation set remains in tact.Please submit your revised manuscript by Nov 13 2024 11:59PM. If you will need more time than this to complete your revisions, please reply to this message or contact the journal office at plosone@plos.org. Please include the following items when submitting your revised manuscript:A rebuttal letter that responds to each point raised by the academic editor and reviewer(s). You should upload this letter as a separate file labeled 'Response to Reviewers'.A marked-up copy of your manuscript that highlights changes made to the original version. You should upload this as a separate file labeled 'Revised Manuscript with Track Changes'.An unmarked version of your revised paper without tracked changes. You should upload this as a separate file labeled 'Manuscript'.

We look forward to receiving your revised manuscript.

Kind regards,

Panos Liatsis, PhD

Academic Editor

PLOS ONE

Journal Requirements:

1. When submitting your revision, we need you to address these additional requirements. Please ensure that your manuscript meets PLOS ONE's style requirements, including those for file naming. The PLOS ONE style templates can be found at https://journals.plos.org/plosone/s/file?id=wjVg/PLOSOne_formatting_sample_main_body.pdf and https://journals.plos.org/plosone/s/file?id=ba62/PLOSOne_formatting_sample_title_authors_affiliations.pdf 2. Please note that PLOS ONE has specific guidelines on code sharing for submissions in which author-generated code underpins the findings in the manuscript. In these cases, we expect all author-generated code to be made available without restrictions upon publication of the work. Please review our guidelines at https://journals.plos.org/plosone/s/materials-and-software-sharing#loc-sharing-code and ensure that your code is shared in a way that follows best practice and facilitates reproducibility and reuse. 3. Thank you for stating the following in the Acknowledgments Section of your manuscript: "The authors would like to acknowledge the support of Princess Nourah bint Abdulrahman University Researchers Supporting Project number (PNURSP2024R435), Princess Nourah bint Abdulrahman University, Riyadh, Saudi Arabia." We note that you have provided funding information that is not currently declared in your Funding Statement. However, funding information should not appear in the Acknowledgments section or other areas of your manuscript. We will only publish funding information present in the Funding Statement section of the online submission form. Please remove any funding-related text from the manuscript and let us know how you would like to update your Funding Statement. Currently, your Funding Statement reads as follows: “The authors received no specific funding for this work.” Please include your amended statements within your cover letter; we will change the online submission form on your behalf. 4. PLOS requires an ORCID iD for the corresponding author in Editorial Manager on papers submitted after December 6th, 2016. Please ensure that you have an ORCID iD and that it is validated in Editorial Manager. To do this, go to ‘Update my Information’ (in the upper left-hand corner of the main menu), and click on the Fetch/Validate link next to the ORCID field. This will take you to the ORCID site and allow you to create a new iD or authenticate a pre-existing iD in Editorial Manager.

Additional Editor Comments :

The work presents an interesting investigation on the use of deep learning methods to the problem of crop productivity classification. However, there are some major issues that the authors need to address. In specific:

a) The novelty of the research is incremental. The authors have combined existing approaches, without clear justification for the choice of the underlying algorithms, and proposed the use of an ensemble approach, i.e., CNN and Bi-GRU. The concern here is a) why these approaches were selected, b) what were the various other approaches that were considered, c) what was the methodology, and experimental results to support the choice, pursued to select the two methods and d) using two underlying systems, one for each modality, does not constitute an ensemble.

b) The research contributions of the work are not related to the novelty and performance, but rather they relate to the impact of the application area, which is indeed common with all other works in this field.

c) The authors should provide a detailed description of the datasets, and the specific challenges associated with each dataset. This will allow them to justify the choice of architectures/methods.

d) The ensemble approach is typically based on using simpler models, which provide for complementary features and decisions, and is characterized by a larger number of models. The larger number of models offers robustness and complementarity demonstrating better performance than that of the underlying models. In this work, there is fusion of two systems, rather than an ensemble, however, there is no substantial evidence (in terms of results) that this leads to better performance than each of the two systems. The recommendation is to consider multiple approaches to extract information from the original image and time series data, which can be combined together into a robust group of features.

e) It is not clear what the specific architectures for the CNN and the Bi-GRU are. What was the process for designing these architectures? The authors should provide information on the design process, including experimental results to support their final architectural recommendations.

f) Based on the description of the work, the dataset has been balanced prior to training and testing. This introduces data leakage where information from samples that should be used in validation/testing only, has been used in training, which impacts on the accuracy of the results and indeed, it's generalization. Thus, the experiments need to be repeated and depending on the training strategy, the dataset should be split into training and testing, followed by balancing the training set only, and testing on the uncontaminated testing set or using k-fold cross-validation, where at each instance, only the folds of the training set are balanced, while the fold of the validation set remains in tact.

Reviewers' comments:

Reviewer's Responses to Questions

**Comments to the Author**

1. Is the manuscript technically sound, and do the data support the conclusions?

Reviewer #1: Yes

Reviewer #2: Yes

2. Has the statistical analysis been performed appropriately and rigorously? 

Reviewer #1: Yes

Reviewer #2: Yes

3. Have the authors made all data underlying the findings in their manuscript fully available?

Reviewer #1: Yes

Reviewer #2: Yes

4. Is the manuscript presented in an intelligible fashion and written in standard English?

Reviewer #1: Yes

Reviewer #2: Yes

5. Review Comments to the Author

Reviewer #1: 1. The integration of Bi-GRU and Time Series CNN for predicting crop yield is a creative and technically sound solution that addresses the complexity of the variables influencing agricultural output. The use of advanced ensemble methods improves accuracy and robustness.

2. The use of advanced preprocessing techniques, such as Adaptive Synthetic Sampling and feature engineering using SVM and NGD, demonstrates a strong methodological foundation.

3. While the mathematical formulations are sound, a clearer narrative explaining how these methods improve crop yield prediction for a broader audience would enhance the paper's accessibility.

4. While the proposed model shows high accuracy, the limitations of deep learning methods (e.g., need for large datasets, computational complexity) should be discussed in more detail.

5. The datasets used in the study are briefly mentioned, but more detailed descriptions of the datasets, especially regarding their specific characteristics and challenges, would help readers understand the complexities the model is addressing.

6. Utilize the following references to improve the introduction section.

i. https://www.doi.org/10.1038/s41598-023-42678-x

ii. Rao, M. V., Sreeraman, Y., Mantena, S. V., Gundu, V., Roja, D., & Vatambeti, R. (2023). Brinjal Crop yield prediction

using Shuffled shepherd optimization algorithm based ACNN-OBDLSTM model in Smart Agriculture . Journal of

Integrated Science and Technology, 12(1), 710.https://pubs.thesciencein.org/journal/index.php/jist/article/view/a710

Reviewer #2: The manuscript presents a well-defined and organized methodology for improving agricultural production prediction. This is achieved by utilizing an integrated ensemble model that combines Bi-GRU and Time Series CNN. The text provides a complete overview of the phases involved in data collection, preprocessing, feature extraction, and model construction.

Approach or systematic procedure The authors include comprehensive information regarding the methodologies utilized, including the use of Support Vector Machine (SVM) for feature selection, Normalized Google Distance (NGD) for correlation analysis, and the Rising Star algorithm for feature ranking. The mathematical expressions and equations underlying the suggested model are clearly articulated, hence improving the clarity and replicability of the study.

Data Availability: The manuscript utilizes three datasets (ECP-DSI, ECP-DSII, and ECP-DSIII), and provides references to their respective sources. Nevertheless, the publication ought to have direct hyperlinks to these datasets or explicitly indicate where the data may be obtained, so improving data accessibility for future researchers.

Data analysis and interpretation:

Examination The data analysis is characterized by its robustness and appropriateness, as it utilizes a range of performance metrics (accuracy, precision, recall, MAE, MSE, and RMSE) to assess the suggested model in comparison to baseline approaches. The comparative study, as shown in Figures 8-11, effectively showcases the model's performance and confirms its superiority over baseline approaches.

Concise and organized display of findings: The results are presented in a clear and concise manner, accompanied by pertinent statistics, tables, and explanations. The book offers a thorough assessment of the model's performance, demonstrating that the suggested model surpasses current methods in accurately predicting crop yields across several datasets.

Ethics and Conflicting Interests:

Ethical Considerations: The research does not present any ethical concerns, and the authors have stated that they have no conflicts of interest. The study excludes the participation of both human and animal subjects, hence reducing ethical problems.

Acknowledgements and financial support Transparency: The authors have explicitly acknowledged the financial support provided by Princess Nourah bint Abdulrahman University, so fulfilling the criteria for transparency in funding and potential conflicts of interest.

Originality and Significance:

The manuscript introduces a noteworthy breakthrough in the realm of agricultural management and crop production prediction. This is achieved by combining advanced deep learning algorithms with ensemble modeling. The suggested method overcomes the constraints of conventional and current machine learning methods, hence providing more precise and resilient predictions.

Wide-ranging Relevance and Applicability: Crop yield prediction is a widely studied subject, especially due to the worldwide concerns around food security. The discoveries have the capacity to influence agricultural practices by facilitating more effective distribution of resources, control of risks, and development of strategic plans.

6. PLOS authors have the option to publish the peer review history of their article (what does this mean?). If published, this will include your full peer review and any attached files.

Reviewer #1: **Yes: **Ramesh Vatambeti

Reviewer #2: No

---

## [Author Response · Author response to Decision Letter 0]

13 Nov 2024

All comments are answered and attached in "Response to the Reviewer Comments" File.

---

## [Decision Letter · Decision Letter 1]

15 Dec 2024

ECP-IEM: Enhancing Seasonal Crop Productivity with Deep Integrated Models

PONE-D-24-34725R1

Dear Dr. Mustafa,

We’re pleased to inform you that your manuscript has been judged scientifically suitable for publication and will be formally accepted for publication once it meets all outstanding technical requirements.

Kind regards,

Panos Liatsis, PhD

Academic Editor

PLOS ONE

Additional Editor Comments (optional):

Reviewers' comments:

Reviewer's Responses to Questions

**Comments to the Author**

1. If the authors have adequately addressed your comments raised in a previous round of review and you feel that this manuscript is now acceptable for publication, you may indicate that here to bypass the “Comments to the Author” section, enter your conflict of interest statement in the “Confidential to Editor” section, and submit your "Accept" recommendation.

Reviewer #1: All comments have been addressed

Reviewer #2: All comments have been addressed

2. Is the manuscript technically sound, and do the data support the conclusions?

Reviewer #1: Yes

Reviewer #2: Yes

3. Has the statistical analysis been performed appropriately and rigorously? 

Reviewer #1: Yes

Reviewer #2: Yes

4. Have the authors made all data underlying the findings in their manuscript fully available?

Reviewer #1: Yes

Reviewer #2: Yes

5. Is the manuscript presented in an intelligible fashion and written in standard English?

Reviewer #1: Yes

Reviewer #2: Yes

6. Review Comments to the Author

Reviewer #1: No further comments. All my comments are properly addressed. Quality of the manuscript has been improved.

Reviewer #2: The manuscript is well-structured, methodologically sound, and presents a novel approach with significant potential impact.

7. PLOS authors have the option to publish the peer review history of their article (what does this mean?). If published, this will include your full peer review and any attached files.

Reviewer #1: No

Reviewer #2: No

---

## [Editor Report · Acceptance letter]

10 Jan 2025

PONE-D-24-34725R1 

PLOS ONE

Dear Dr. Mustafa, 

I'm pleased to inform you that your manuscript has been deemed suitable for publication in PLOS ONE. Congratulations! Your manuscript is now being handed over to our production team.

Kind regards, 

on behalf of

Professor Panos Liatsis 

Academic Editor

PLOS ONE